# Construction of a Compact and High-Precision Classifier in the Inductive Learning Method for Prediction and Diagnostic Problems

**Roman Kuzmich** [1,2,*], **Alena Stupina** [1,3], **Andrey Yasinskiy** [1,2] , **Mariia Pokushko** [1,2,4], **Roman Tsarev** [5,6] and **Ivan Boubriak** [7]

1    Department of Business Informatics and Business Process Modeling, Siberian Federal University, Krasnoyarsk 660041, Russia
2    Scientific Center for Information Technology and Artificial Intelligence, Sirius University of Science and Technology, Sochi 354349, Russia
3    Department of Systems Analysis and Operations Research, Reshetnev Siberian State University of Science and Technology, Krasnoyarsk 660037, Russia
4    Superior Engineering School, University of Cadiz, 11519 Cadiz, Spain
5    Department of Applied Mathematics, MIREA—Russian Technological University (RTU MIREA), Moscow 119454, Russia
6    V.A. Trapeznikov Institute of Control Sciences of Russian Academy of Sciences, Moscow 117997, Russia
7    Sir William Dunn School of Pathology, Oxford University, South Parks Road, Oxford OX1 3QU, UK
*    Correspondence: rkuzmich@sfu-kras.ru

**Abstract:** The study is dictated by the need to make reasonable decisions in the classification of observations, for example, in the problems of medical prediction and diagnostics. Today, as part of the digitalization in healthcare, decision-making by a doctor is carried out using intelligent information systems. The introduction of such systems contributes to the implementation of policies aimed at ensuring sustainable development in the health sector. The paper discusses the method of inductive learning, which can be the algorithmic basis of such systems. In order to build a compact and high-precision classifier for the studied method, it is necessary to obtain a set of informative patterns and to create a method for building a classifier with high generalizing ability from this set of patterns. Three optimization models for the building of informative patterns have been developed, which are based on different concepts. Additionally, two algorithmic procedures have been developed that are used to obtain a compact and high-precision classifier. Experimental studies were carried out on the problems of medical prediction and diagnostics, aimed at finding the best optimization model for the building of informative pattern and at proving the effectiveness of the developed algorithmic procedures.

**Keywords:** pattern; optimization model; classifier; boosting criterion; prediction and diagnostic problems

## 1. Introduction

The problems of medical prediction and diagnostics are very important from the point of view of preserving life and health of society. Good health of society is one of the goals of its sustainable development, according to the concept of sustainable development. Sustainable development in the health sector contributes to the dynamic development of society in other areas. It should be noted that the key advantage of diagnosing a disease and predicting its outcome at an early stage is the rapid appointment of the necessary therapy to the patient to save his life. In order to help the doctor, make a correct diagnosis, and make a prediction for the development of the disease, decision support systems based on machine learning methods are being actively developed today. Most machine learning methods (for example, artificial neural networks) are aimed only at high classification accuracy in diagnosing and predicting a disease. However, when solving the problems of

medical prediction and diagnostics, first of all, questions arise related to the interpretability of the obtained results and the validity of the proposed solutions. Therefore, the decision support system used for the problems of medical prediction and diagnostics must justify the proposed solutions and interpret the result.

The basis of such a system should be classification methods, which, in addition to the solution itself, explicitly represent the decision rule obtained as a result of revealing knowledge from the initial data of the problem. These methods include methods of inductive learning. From the family of inductive learning methods, we single out logical classification algorithms, the distinguishing feature of which is the formalization of the identified knowledge in the form of a set of rules, i.e., a set of patterns described by a simple logical formula. It was shown in [1] that algorithms in which the choice of a classification decision is based on a finite set of rules are very promising in terms of interpretability.

Give a brief description of the main logical classification algorithms, their advantages, and disadvantages, which help to justify the prospects of using the method of logical analysis of data for the problems of medical prediction and diagnostics.

Klivans et al. [2] proposed a simple algorithm called "Decision list". The essence of the algorithm is that, for each class, a set of patterns is built. The totality for the patterns of all classes makes up the decision list. When classifying a new observation, patterns are checked sequentially until one of them covers the observation, defining its class. If the observation does not cover any pattern, the classification is rejected. One solution in this case is to assign such an observation to the class that has the minimum price of a mistake.

Note that, at each iteration of the algorithm for constructing the decision list, exactly one pattern is built, which covers the maximum number of observations in the training sample of its class and the minimum number of observations from other classes. After the rule is built, the observations that it has covered are removed, and the algorithm proceeds to search for the next rule based on the remaining observations in the training sample until the number of remaining observations becomes less than the value specified by the researcher.

The advantages of the decision list include: the interpretability of the rules and the ability to handle data with gaps. The biggest disadvantage is the fact that each observation is classified by only one rule, which prevents the rules from compensating for each other's mistakes.

Loh [3] presented the "Decision tree" algorithm—a logical classification algorithm that differs from the decision list in that all rules are built simultaneously during synthesis. The advantages of decision trees are the absence of classification failures and the simplicity and interpretability of the classification results. Besides the fact that each observation is classified by only one rule, the main disadvantage for a decision tree is the non-optimality of the "greedy" tree increasing strategy. To solve this problem, various heuristic techniques are used, in particular, reduction.

Algorithms for simple and weighted rule voting. Let $\varphi : U \rightarrow \{0, 1\}$, which is a predicate defined on a set of observations $U$. The predicate $\varphi$ covers an observation $u$ if $\varphi(u) = 1$. A predicate is called a pattern if it covers a sufficiently large number of observations in the same class and practically does not cover observation in other classes. For each class $k \in Y$, a set of patterns necessary for classifying observations in this class is constructed: $R_k = \left\{ \varphi_k^t : U \rightarrow \{0, 1\} \mid t = 1, \ldots, T_k \right\}$. It should be noted that if $\varphi_k^t(u) = 1$, then the $\varphi_k^t$ rule assigns the observation $u \in U$ to the class $k$. If $\varphi_k^t(u) = 0$, then the $\varphi_k^t$ rule refrains from classifying the observation $u$.

Vorontsov [4] proposes a simple voting algorithm that calculates the proportion of rules in the sets $R_k$ that refer observation u to each of the classes:

$$W_k(u) = \frac{1}{T_k} \sum_{t=1}^{T_k} \varphi_k^t(u),$$

and determines the observation $u$ to the class with the largest share of votes:

$$a(u) = \arg_{k \in Y} max W_k(u).$$

If the share of votes is the same for several classes, the one is selected for which the price of mistake is less. The normalizing multiplier $1/T_k$ is introduced so that sets with a different number of rules have the same influence on decision making when classifying a new observation.

Vorontsov [4] also proposes a weighted voting algorithm that takes into account the possibility of rules having different values. Therefore, each rule $\varphi_k^t$ is assigned a weight $\alpha_k^t \geq 0$ and, during voting, the weighted sum of votes is taken:

$$W_k(u) = \frac{1}{T_k} \sum_{t=1}^{T_k} \alpha_k^t \varphi_k^t(u).$$

The weights are normalized to unity: $\sum_{t=1}^{T_k} \alpha_k^t = 1$, for all $k \in Y$. Based on this, simple voting is a special case of weighted voting, when the weights are the same and equal to $1/T_k$.

Weinzweig [5] proposed the KORA simple voting algorithm, which uses strategy for pattern enumeration (the depth-first search), and Lbov [6] proposed the TEMP algorithm (breadth-first search).

The breadth-first search algorithm is based on the following heuristic assumptions:

1.  The set of terms is chosen so well that there are already a sufficient number of informative rules among the rules with degree 2 or 3.
2.  To search for rules, you can apply an exhaustive search, since their degree is limited from above by the number 3.

This algorithm generates its own list of rules $R_k$ for each class $k \in Y$. Each rule contains no more than three terms selected from a set of predicates. The basis of the algorithm is a recursive procedure for increasing rules, which adds terms to the rule by any means and enters into the list of rules only the best rules that satisfy the selection criteria $D_k(\varphi) \geq D_{min}$ and $E_k(\varphi) \leq E_{max}$, where $D_k(\varphi)$ is proportion of covered positive observations, and $E_k(\varphi)$ is the proportion of negative observations among all covered.

The number of received rules is strongly influenced by the values of the parameters $D_{min}$ and $E_{max}$. If the selection criteria are set too strong, the algorithm may not find any rules at all. Otherwise, when the criteria are too weak, the algorithm will spend time on exhaustive search and evaluating a large number of uninformative rules. In practice, the values of these parameters are selected experimentally.

The essence of the breadth-first search algorithm is as follows. The process of searching for patterns begins with the construction of rules with a degree equal to 1. To do this, no more than $T_1$ of the most informative terms are selected from the base set of terms $B$. We obtain rules consisting of one term. Then one term from the remaining set of terms $B$ is added to each of them in all possible ways. As a result, we obtain no more $T_1|B|$ rules with degree 2, from which $T_1$'s most informative rules are again selected, and so on. At each step of the process, an attempt is made to add one term to each of the existing rules. Rule increase stops either when the maximum degree $K$ is reached, or when none of the rules can be improved by adding a term.

As a result, the classifier can consist of rules with varying degrees, since the best rules collected from all steps of the breadth-first search algorithm fall into it.

A compromise between the quality and speed of the algorithm is provided by the parameter $T_1$. With $T_1$ equal to 1, the breadth-first search algorithm is extremely fast and builds a single rule by adding terms one by one. In fact, it is the same as the greedy algorithm. With an increase in $T_1$, the search space expands and the algorithm starts to work more slowly, but it finds more informative rules. For $T_1$ equal to $\infty$, the algorithm will perform an exhaustive search. In practice, the maximum value of the parameter $T_1$ is chosen, at which point the search takes an acceptable time.

The disadvantage of the breadth-first search algorithm is a greedy search strategy—terms are optimized separately, and when selecting each term, only previous terms are

considered, but not subsequent terms. The advantage of the breadth-first search algorithm is its higher speed compared to the depth-first search algorithm.

Additionally, the KORA and TEMP algorithms have a common disadvantage: they do not tend to increase the difference in patterns. This disadvantage can be eliminated within the framework of the method of logical analysis of data by purposefully developing an optimization model for building patterns, which helps in increasing the difference between the rules in the classifier.

## 2. Materials and Methods

### 2.1. Formulation of the Problem

The paper focuses on the method of logical analysis of data, which is used to solve problems of binary classification [7]. It should be noted that the method was previously successfully tested on a number of practical problems [8–12].

Present the initial formulation for the problem of binary classification. The data sample consists of disjoint sets $\Psi^+$ (set of observations in the positive class) and $\Psi^-$ (set of observations in the negative class) of n-dimensional vectors. The elements of the vector take different types of values (binary, nominal, and quantitative).

The essence of solving such problems is to determine the class label for a new observation, to which it can be assigned. The definition of the class label is carried out on the basis of a classifier consisting of patterns extracted from the original data sample.

Note that each pattern in the classifier is a conjunction of attributes or their negations. Its properties are coverage (number of captured observations in its class) and degree (number of terms from which it is composed). A feature of conjunctions for a small degree is that they are visual and easy to perceive, since they appear to be logical statements familiar to humans.

Any pattern covers only a part of the observations from the set *U*. By making up a composition from a certain number of patterns, you can obtain a classifier that can classify any observations from the sample. The decision to classify a new observation is made according to the simple voting algorithm or according to the weighted voting algorithm. The weights of patterns covering a new observation will be equal when deciding (simple voting) or will be determined considering the informativity of the pattern (weighted voting).

The quality of the formed rule is determined using the concept of informativity. The concept of informativity is defined through the property "pattern coverage". When the informativity of the pattern is higher, the more it covers the observations in the class on the basis of which it is built, and the less it covers the observations in another class. Based on the definition, two types of patterns are distinguished: pure patterns, which do not cover observations in another class, and partial patterns, which are helped to cover a limited number of observations in another class.

Based on the concept "informativity of a pattern", two criteria must be observed to construct informative rules: maximizing the coverage of observations in "own" class ($p$) and minimizing the coverage of observations in another class ($n$). It follows that patterns are less useful from a classification point of view if they cover too few observations or cover observations in the positive and negative classes in approximately the same proportion as they were represented in the entire sample.

Thus, in order to build patterns, it is necessary to compose such optimization models, which are based on the idea of convolution of these two criteria into one criterion. Furnkranz and Flach [13] show that it is not a trivial problem to propose an adequate convolution of two criteria within a single optimization model.

In this paper, we propose three concepts of optimization models for the building of patterns: the maximum coverage of observations in "own" class, the coverage of significantly different subsets of observations in the training sample, and the use of the boosting criterion as the objective function of the optimization model. The objective function and the constraint function in the developed optimization models should include both criteria for the informativity of a pattern. It is necessary to determine the best of the developed

optimization models for the building of informative pattern by comparative analysis on practical problems. At the next stage of the method, in order to make a decision on the classification of a new observation, it is necessary to build a compact classifier with a high generalizing ability based on the obtained patterns. For this purpose, algorithmic procedures aimed at reducing the number of patterns in the classifier, while maintaining its generalizing ability, have been developed and investigated in the paper.

### 2.2. Data Preparation and Feature Selection

The method studied in the paper consists of a number of steps aimed at extracting as much information as possible from the initial data, which are necessary to make a decision about whether an observation belongs to a particular class.

The first step of the method is to check for the type of attributes present in the original data sample. A certain limitation of the method is that it works only with binary attributes. If not all attributes are binary, then a binarization procedure is necessary. Masich [14] proposed a simple attribute binarization method, called a single binarization method. The essence of this method is that each quantitative variable is associated with several binary variables. The value of a binary variable is 1 if the value of the corresponding quantitative variable is above a certain threshold, and vice versa. Initially, the number of binarization thresholds for a feature is set by the researcher independently. Vorontsov [4] proposes a procedure for reducing the number of binarization thresholds for a feature. It is based on an increase in the maximum gain in informativity when the observation zones of positive and negative classes merge. Reducing the number of binarization thresholds for features directly leads to a reduction in the number of binary variables involved in the classification.

The next step of the method is the construction of a reference set of attributes, so, the problem of attribute selection is solved, which will be used in the future when building patterns. Hammer and Bonates [11] propose an approach that consists in removing redundant attributes from the original sample and determining some subset $S$, which will help to solve the classification problem with high accuracy. In what follows, when working with the method, the projections $\Psi_S^+$ and $\Psi_S^-$ of the sets $\Psi^+$ and $\Psi^-$ onto $S$ will be used. This approach is based on the concept of the minimum support set of attributes $S$, so, the minimum set of variables that helps one to separate $\Psi^+$ and $\Psi^-$. The search for a reference set of attributes $S$ is carried out in the form of a combinatorial optimization problem, where the set of attributes is minimized in the objective function, and the intersection of $\Psi^+$ and $\Psi^-$ is not helped in the constraint function.

Formulate the problem of finding the minimum support set of features $S$ in the form of a combinatorial optimization problem. Assign to each feature $u_i (i = 1, \dots, t)$ of the original sample a new binary variable $r_i$. Values $r_i$ equal to 0 indicate that they do not belong to $S$, and values of $r_i$ equal to 1 belong to $S$. We introduce the notation $Q = (q_1, q_2, \dots q_t)$ : it is the vector associated with $\Psi_S^+$, $P = (p_1, p_2, \dots p_t)$ : it is the vector associated with $\Psi_S^-$. Enter variable:

$$m_i(Q, P) = \begin{cases} 1, q_i \neq p_i, \\ 0, q_i = p_i. \end{cases}$$

The condition of disjoint sets $\Psi_S^+$ and $\Psi_S^-$ requires that the inequality $\sum m_i(Q, P)r_i \geq 1$ hold for any $Q \in \Psi_S^+$ and $P \in \Psi_S^-$. It should be emphasized that, in order to strengthen the limitation of the optimization model, the number 1 on the right side of the inequality can be replaced by an integer $g$. As a result, we obtain the following optimization model for searching a reference set of features:

$$\sum_{i=1}^{t} r_i \to min,$$

$$m_i(Q, P)r_i \geq g \quad for \ any \quad Q \in \Psi_S^+, \ P \in \Psi_S^-, \ r \in \{0,1\}^t$$

Kuzmich et al. [15] offers an alternative way to form a reference set of features on the basis of developed algorithmic procedure based on the evaluating the importance of features. The importance of a feature is understood as the frequency of its inclusion in the patterns presented in the classifier. Based on the definition, it can be established that the number of feature inclusions in the patterns is directly proportional to its importance. The fewer times a feature is involved in the formed patterns, the less important it is when deciding on the classification of observations. Such a feature is a candidate for removal from the original set of features.

After the formation of the reference set of attributes, we proceed directly to the building of patterns. Pattern is the basic element of the studied method. Patterns are obtained based on all observations in the training sample for each class. If they are formed for $\Psi_S^+$, then they are positive, if for $\Psi_S^-$, then they are negative.

### 2.3. Optimization Model for Building Patterns with Maximum Coverage

Consider the building a set of patterns for observations in a positive class. For each observation $\alpha \in \Psi_S^+$, we will search a $\alpha$-pattern that help to cover the maximum number of observations in the set $\Psi_S^+$. To do this, we set the desired logical rule using binary variables $X = (x_1, x_2, \ldots, x_t)$:

$$x_k = \begin{cases} 1, & if\ the\ k-th\ attribute\ is\ in\ the\ rule. \\ 0, & otherwise. \end{cases}$$

The objective function of the optimization model is the total number of observations $\Psi_S^+$ for the $\alpha$-pattern. The following rule is used to determine the value of the coverage by the $\alpha$-pattern: the observation $\sigma \in \Psi_S^+$ is included in the formed pattern when it differs from $\alpha \in \Psi_S^+$ only in those attributes that do not participate in the building of the pattern.

The constraint function on the coverage of observations $\Psi_S^-$ acts as a constraint of the optimization model. When building pure patterns, the $\alpha$-pattern should not cover a single observation $\Psi_S^-$, and when building partial patterns, the $\alpha$-pattern can cover a certain number of observations $\Psi_S^-$.

As a result, in [16–19] an optimization model for the building of patterns is given:

$$\sum_{\sigma \epsilon \Psi_S^+} \prod_{\substack{k=1 \\ \sigma_k \neq \alpha_k}}^{t} (1 - x_k) \to max, \tag{1}$$

$$\sum_{\substack{k=1 \\ \beta_k \neq \alpha_k}}^{t} x_k \geq g \quad for\ any\ \beta \epsilon \Psi_S^-,\ x \epsilon \{0,1\}^t, \tag{2}$$

where $g$ is integer positive number.

According to the optimization model (1)–(2), pure patterns are built for observations in the positive class, which do not capture any observation $\Psi_S^-$. Similarly, pure patterns are formed for observations in the negative class.

Samples in real classification problems are characterized by the presence of missing data and outliers, which leads to the problem of inseparability for observations in positive and negative classes. Consequently, the optimization model reacts to this fact by increasing the degree (the number of variables in the formed patterns) and reducing the coverage of the resulting patterns. As a result, the possibility of constructing a high-precision classifier with interpretable patterns is sharply reduced.

In this case, a transition to the building of partial patterns is proposed, so patterns that can cover a certain number of observations in the opposite class. Practical studies show that the degree of partial patterns is usually less than that of pure patterns, and the coverage is greater [16].

The transition to partial patterns is carried out by weakening the constraint (2) in the optimization model (1)–(2) [20]:

$$\sum_{\beta \epsilon \Psi_S^-} Z_\beta \leq G, \tag{3}$$

where

$$Z_\beta = \begin{cases} 0, \; if \; \sum\limits_{\substack{k=1 \\ \beta_k \neq \alpha_k}}^{t} x_k \geq g; \\ 1, \; otherwise. \end{cases}$$

where $G$ is the number of opposite class observations that can be covered by a pattern.

According to model (1)–(3), patterns with the greatest coverage are obtained. As a rule, such patterns have a small degree, i.e., consist of a small number of terms and correspond to large areas in the feature space. They are located to cover observations of the opposite class, which negatively affects their informativity. Therefore, in order to increase the informativity of the patterns, an algorithmic procedure for their growth is developed. It is applied to each constructed pattern. The meaning of the procedure is to maximize the degree of patterns while maintaining coverage [16]:

$$\sum_{j=1}^{t} x_j \rightarrow max, \; fc(X) = fc'(X),$$

where $fc(X)$ is the coverage value for the pattern before the algorithmic increasing procedure, $fc'(X)$ is the coverage value for the pattern after the algorithmic increasing procedure.

As a result of applying the increasing procedure, we will obtain patterns with maximum coverage and with a higher degree, thereby increasing the reliability of the decisions made by the classifier. Increasing the reliability of solutions is based on increasing the informativity of patterns, since the number of observations in the "own" class (the value of the objective function for the pattern) remains unchanged, while the number of observations in another class captured by the rule decreases.

### 2.4. Optimization Model for the Building of Patterns Based on the Boosting Criterion

In the optimization models (1)–(2) and (1)–(3), the objective Function (1) maximizes the coverage of observations in "own" class ($p$), and the constraint Functions (2) and (3) set an upper limit on the number of observations in another class ($n$).

Furnkranz and Flach [13] introduced the boosting criterion, which is proposed in this optimization model as a convolution of two initial criteria ($p \rightarrow max$ and $n \rightarrow min$). As a result, the objective function of the optimization model for the building of patterns appear as such:

$$\sqrt{p} - \sqrt{n} \rightarrow max. \tag{4}$$

It should be noted that in the objective function an attempt is made to simultaneously maximize the number of observations in "own" class and minimize the number of observations in another class. The basis for the adequacy of the boosting criterion is the fact that it has been successfully used for a long time in the algorithm of the same name, which belongs to the weighted rules voting algorithm. In the method of logical analysis of data, the boosting criterion is used for the first time to obtain patterns.

We will use (3) as a constraint function in the optimization model. It will help to limit the number of covered observations in another class.

### 2.5. Optimization Model for the Building of Patterns with Coverage of Significantly Different Subsets of Observations in the Training Sample

The idea that underlies this optimization model is that patterns in the classifier must be different or unique, otherwise they become useless in classification.

According to the objective Function (1), each pattern is formed with the maximum coverage of observations of "own" class, while capturing those observations that are typical

representatives of the class. In turn, atypical observations of the class remain uncovered, i.e., there are no patterns in the classifier that take them into account. As a result of such an optimization model, we obtain a set of similar patterns for the class, thereby reducing the quality of the classification.

It is proposed to make changes to the objective Function (1) to obtain a classifier with a higher rule difference, which is built on significantly different subsets of observations [21]:

$$\sum_{\sigma \epsilon \Psi_S^+} K_\sigma \times \prod_{\substack{k=1 \\ \sigma_k \neq \alpha_k}}^{t} (1 - x_k) \rightarrow max. \tag{5}$$

where $K_\sigma$ is the weight of the observation $\sigma \epsilon \Psi_S^+$, which decreases with coverage $\sigma$ and affects the priority of $\sigma$ participation in the formation of subsequent patterns.

In the practical application of the optimization model with the objective Function (5), it is necessary to perform two actions. First, set initial weights for all observations. Secondly, set the rule for changing the weights for the observations that took part in the formation of the current pattern. The initial weights for all observations are proposed to be set equal to 1. The rule for changing the weight for each observation that took part in the formation of the current pattern:

$$K_{i+1} = max\left[0, K_i - \frac{1}{N_{max}}\right],$$

where $K_i$, $K_{i+1}$ are the weights of the covered observation in the formation of the current and next patterns, and $N_{max}$ is the maximum number of observation participations in the formation of patterns (set by the researcher).

As a result, various patterns are obtained on the basis of the optimization model with objective Function (5). At the stage of constructing a classifier, those patterns are selected from them for which the values of the objective function are greater than zero.

### 2.6. Algorithm for Deciding on the Classification of a New Observation

After the building of all patterns for each class, we proceed to the next step of the method. It consists in constructing a compact classifier with a high generalizing ability from the found patterns.

The basic concept of constructing a classifier is the idea of combining all the found patterns. Since the pattern is based on the observation in the training sample, their total number in the classifier is equal to the power of the training sample. Therefore, with a large amount of training samples in the original problem, it is necessary to develop algorithmic procedures aimed at reducing the number of patterns in the classifier while maintaining its generalizing ability.

The paper presents two algorithmic procedures used to obtain a compact classifier with high generalizing ability.

The idea of the first algorithmic procedure is connected with the change in basic observations for the formation of patterns [22]. If, in the original version of the method, the initial observations in the training sample were used as basic observations, then the algorithmic procedure suggests using centroids obtained on the basis of the initial observations using the "k-means" algorithm [23]. In this case, the use of the "k-means" algorithm is justified by the fact that the resulting set of centroids optimally represents the distribution of observations in the training sample.

The developed algorithmic procedure belongs to the procedures for speeding up the search for rules, since even before the stage of their formation, it helps to reduce their number, thereby reducing the complexity of building a classifier.

Thus, as a result of the developed algorithmic procedure, we obtain a classifier whose number of rules is equal to the total number of centroids for all classes. An experimentally tuned parameter of the algorithmic procedure that affects the classification accuracy is the number of centroids for each class.

Consider another algorithmic procedure used to obtain a compact classifier with high generalizing ability. It helps to reduce the classifier based on assessing the informativity of its patterns [24].

In this study, to measure the informativity of the patterns, it is proposed to use the normalized boosting criterion since it adequately evaluates the informativity of the pattern and is easy to calculate:

$$H(p,n) = \sqrt{p/P} - \sqrt{n/N}, \tag{6}$$

where $p$ is the number of observations in "own" class captured by the pattern; $P$ is the total number of observations in its class in the training sample; $n$ is the number of observations in another class captured by the pattern; and $N$ is the total number of observations in another class in the training sample.

The starting point is a classifier consisting of a complete set of patterns. The informativity of each pattern must be assessed on the basis of Formula (6). Rank the patterns in descending order of the informativity value for each class. It should be noted that patterns with a low informativity value are not statistically reliable—there are too many of them that make more mistakes on the observations in the examining sample than on the observations in the training sample. Therefore, it is necessary to perform the procedure of reducing the classifier by excluding patterns from it with a low informativity value.

The criteria for stopping the classifier reduction procedure can be the following parameters set by the researcher: the minimum number of patterns for the class and the number of uncovered observations in the examining sample. After all, an increase in the number of uncovered observations directly indicates the insufficiency of the patterns' number in the classifier.

Since the most uninformative patterns are excluded from the classifier, the accuracy of the classification is preserved or changes slightly in one direction or another. The result of such an algorithmic procedure will be a compact classifier with a high generalizing ability.

After building a classifier, it is necessary to make an informed decision on the basis of which a new observation belongs to a particular class. The rationale for the classification decision is the fact that a new observation is covered by the patterns of a particular class.

To classify a new observation, we use the following decision-making algorithm [25]:

1. If an observation is covered by one or more positive patterns and it is not covered by any of the negative patterns, then it is classified as positive.
2. If an observation is covered by one or more negative patterns and it is not covered by any of the positive ones, then it is classified as negative.
3. Choice of voting algorithm:

- Simple voting algorithm. If an observation is covered by $p'$ out of $p$ by positive patterns and $q'$ out of $q$ by negative patterns, then the observation sign is defined as $p'/p - q'/q$.
- Weighted voting algorithm. If an observation is covered by $p'$ out of $p$ by positive patterns and $q'$ out of $q$ by negative patterns, then the observation sign is defined as $\sum_{n=1}^{p'} a_n - \sum_{n=1}^{q'} b_n$, where $a$ and $b$ are weights for positive and negative patterns.

The weight for the $n$-th positive pattern is calculated by the formula:

$$a_n = \frac{H_n}{\sum_{n=1}^{p} H_n},$$

where $H_n$ is the informativity of the n-th positive pattern, which is calculated by the normalized boosting criterion (5).

The sum of the weights of all positive patterns is equal to one: $\sum_{n=1}^{p} a_n = 1$. Similarly, informativity and weight are calculated for the $n$-th negative pattern.

4. If an observation is not covered by any pattern, then it is assigned the class label with the lowest price of mistake.

## 3. Results

We will conduct numerical studies on the problems of medical prediction and diagnostics presented in the UCI: the problem of breast cancer diagnosis and the problem of predicting the outcome of a patient with hepatitis [26]. Numerical studies are carried out in order to determine the best optimization model for the building of informative pattern. All studies are carried out using an intelligent software system developed by the authors [27]. The block diagram of the program operation corresponds to the steps of the method of logical analysis of data described above. The block diagram is shown in Figure 1.

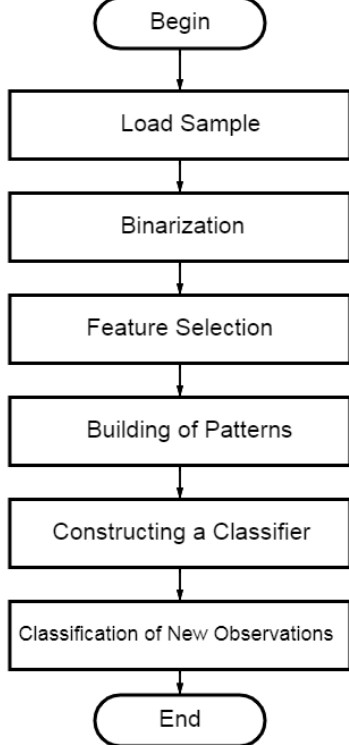

**Figure 1.** The block diagram of the program operation.

The aim of solving the first problem is to recognize observations in two classes based on gene expression values: patients with a malignant tumor and patients with a benign tumor. The aim of solving the second task is to predict the outcome of the disease for patients with hepatitis: death or survival.

Three optimization models were used to find the rules: a model for the building of partial patterns with maximum coverage; a model for the building of partial patterns based on the boosting criterion; and a model for the building pattern with coverage of significantly different subsets of observations in the training set. Note that the objective and constraint functions are specified algorithmically in these optimization models. They are also unimodal and monotonic. To solve optimization problems with such functions, special search optimization algorithms are used [28–30].

In all presented practical problems, 80% of the sample was used for training and 20% of the sample was used to test the classifier.

The classification results for the considered problems are presented below.

Problem 1. Breast cancer diagnosis. For the experiments, a data sample was used, consisting of 212 observations of the positive class (patients with a malignant tumor) and 357 observations of the negative class (patients with a benign tumor). Each observation was described by a vector of 30 different types of features. Two testing methods are given in this paper: percentage split (PS) of the sample and five-fold cross-validation (CV) [31,32]. With percentage split in the problem, 80% of the sample was used as a training sample,

and 20% of the sample was used as an examination sample. The time costs associated with building patterns are also given. The test results are presented in Table 1.

**Table 1.** Classification results for the problem of breast cancer diagnosis.

| Optimization Problem | Set of Rules | Number of Rules | Average Degree of Rule | Cover of Negative Patterns | Cover of Positive Patterns | Classification Accuracy (PS), % | Classification Accuracy (CV), % | Time, s |
|---|---|---|---|---|---|---|---|---|
| Objective Function | Neg. | 269 | 3 | 233 | 8 | 90 | 92 | 11.47 |
| (1), constraint (3) | Pos. | 186 | 3 | 9 | 153 | 96 | 93 | 7.43 |
| Objective Function | Neg. | 269 | 2 | 226 | 6 | 94 | 94 | 16.92 |
| (4), constraint (3) | Pos. | 186 | 2 | 3 | 139 | 96 | 96 | 13.3 |
| Objective Function | Neg. | 24 | 3 | 229 | 8 | 91 | 94 | 12.08 |
| (5), constraint (3) | Pos. | 26 | 3 | 8 | 144 | 96 | 95 | 7.9 |

Problem 2. Predicting the outcome of a patient's disease with hepatitis. For the experiments, a data sample was used, consisting of 32 observations of the positive class (patients with a death outcome) and 123 observations of the negative class (surviving patients). Each observation was described by a vector of 20 different types of features. Two testing methods are given in this paper: percentage split (PS) of the sample and five-fold cross-validation (CV). With percentage split in the problem, 85% of the sample was used as a training sample, and 15% of the sample was used as an examination sample. The time costs associated with building patterns are also given. The test results are presented in Table 2.

**Table 2.** Classification results for the problem of predicting the outcome of a patient' disease with hepatitis.

| Optimization Problem | Set of Rules | Number of Rules | Average Degree of Rule | Cover of Negative Patterns | Cover of Positive Patterns | Classification Accuracy (PS), % | Classification Accuracy (CV), % | Time, s |
|---|---|---|---|---|---|---|---|---|
| Objective Function | Neg. | 123 | 1 | 89 | 8 | 79 | 82 | 5 |
| (1), constraint (3) | Pos. | 23 | 4 | 9 | 11 | 67 | 64 | 1 |
| Objective Function | Neg. | 123 | 1 | 86 | 7 | 86 | 83 | 6.5 |
| (4), constraint (3) | Pos. | 23 | 2 | 4 | 8 | 89 | 77 | 1.5 |
| Objective Function | Neg. | 21 | 1 | 87 | 8 | 79 | 82 | 5.5 |
| (5), constraint (3) | Pos. | 20 | 3 | 9 | 10 | 67 | 62 | 1.2 |

Examples of the patterns obtained for the problem of predicting a patient's disease with hepatitis are shown in Figure 2. Each observation of the examining sample can be covered by several patterns of a positive and negative class at once. The decision on whether a new observation belongs to a particular class is made according to a simple voting algorithm given in this paper.

```
========================= Negative patterns =========================
Pattern №1   (42 <   AGE) (SPIDERS = 0)
Pattern №2   (30 <   AGE) (LIVER_FIR = 0) (0.9 <  BILIRUBIN) (82 <   ALK_PHOSPH)
Pattern №3   (30 <   AGE) (SEX = 0) (ANTIVIRALS = 1) (MALAISE = 0) (SGOT   < 120)
Pattern №4   (36 < 51 AGE) (AGE 36 < 51) (FATIGUE = 0) (ANOREXIA = 1) (LIVER_BIG = 1)
Pattern №5   (30 <   AGE) (SEX = 0) (ANTIVIRALS = 1) (SPIDERS = 0)
Pattern №6   (SEX = 0) (STEROID = 0) (1 <   BILIRUBIN) (82 <   ALK_PHOSPH)
Pattern №7   (SPLEEN_PAL = 0) (SPIDERS = 0)
Pattern №8   (42 <   AGE) (FATIGUE = 0) (HISTOLOGY = 1)

========================= Positive patterns =========================
Pattern №1   (BILIRUBIN  < 1.6)
Pattern №2   (ASCITES = 1) (0.7 <  BILIRUBIN)
Pattern №3   (SPLEEN_PAL = 1) (VARICES = 1)
Pattern №4   (SPIDERS = 1)
Pattern №5   (3.3 <  ALBUMIN)
Pattern №6   (AGE   < 51) (ASCITES = 1)
Pattern №7   (HISTOLOGY = 0)
Pattern №8   (VARICES = 1) (SGOT   < 120)
```

**Figure 2.** Examples of patterns for problem 2.

According to the obtained results on the accuracy of classification for the problems being solved (Tables 1 and 2), the best optimization model for the building of informative patterns is the model for the building of partial patterns based on the boosting criterion.

The classifier, consisting of patterns built on the basis of the optimization model (4)–(3), has a high generalizing ability. It should also be noted that the average degree of the formed rule sets based on this model is lower than that of the formed rule sets based on another optimization models. Consequently, the rules obtained on the basis of the optimization model (4)–(3) are simpler and clearer.

Note that the use of an optimization model with an objective Function (5) to search for patterns makes it possible to make the classifier compact, reducing the number of patterns in it, since rules only with a value of the objective function greater than zero obtain into the classifier.

For the problems being solved, we will test two algorithmic procedures used to obtain a compact classifier with a high generalizing ability.

First, we will conduct experimental studies of the algorithmic procedure, where centroids obtained using the "k-means" algorithm are used as basic observations. To generate centroids for each practical problem, we will use the WEKA program, which provides the user with the ability to preprocess data, solve problems of classification, clustering, regression, search for association rules, as well as visualize data and results [33].

For problem 1, 20 centroids are generated for the set of observations of positive and negative classes. For problem 2, eight centroids are generated for the set of observations of the positive class and 15 centroids for the set of observations of the negative class. The number of observations of the examining sample is 20% for problem 1 and 15% for problem 2 of the total number of observations for the practical problems being solved. In the process of conducting numerous experiments, the number of observations of another class was selected, which can capture the pattern (parameter G in the constraint Function (3)). The results are shown in Tables 3 and 4.

According to Tables 3 and 4, we received a slight change in the classification accuracy for the problems being solved and a reduction in the number of rules in the classifier by 11 times for the problem of breast cancer diagnosis, and for the problem of predicting the outcome of a patient' disease with hepatitis, by six times. Thus, the modification of the method associated with the algorithmic procedure, where centroids are used as basic observations, is effective from the point of view of applicability for constructing rules that form a compact classifier with high generalizing ability.

The next algorithmic procedure for conducting experimental research is the reduction of the classifier, based on the assessment of the informativity of its patterns. The number of observations of the examining sample is 20% for problem 1 and 15% for problem 2 of the total number of observations for the practical problems being solved.

**Table 3.** Comparison of the classifier accuracy with different baseline observations to build patterns (problem of breast cancer diagnosis).

| Set of Rules | Coverage Neg. Observations in the New/Original Classifiers | Coverage Pos. Observations in the New/Original Classifiers | Degree of Rule in the New/Original Classifiers | Number of Rules in the New/Original Classifiers | New Classifier Accuracy, % | The Number of Correctly Classified Observations by the New Classifier, % | Initial Classifier Accuracy, % | The Number of Correctly Classified Observations by the Initial Classifier, % |
|---|---|---|---|---|---|---|---|---|
| Neg. | 253/233 | 9/8 | 2/3 | 20/269 | 90 | 91.2 | 90 | 91.2 |
| Pos. | 9/9 | 174/153 | 3/3 | 20/186 | 96 | | 96 | |

**Table 4.** Comparison of the classifier accuracy with different baseline observations to build patterns (the problem of predicting the outcome of a patient' disease with hepatitis).

| Set of Rules | Coverage Neg. Observations in the New/Original Classifiers | Coverage Pos. Observations in the New/Original Classifiers | Degree of Rule in the New/Original Classifiers | Number of Rules in the New/Original Classifiers | New Classifier Accuracy, % | The Number of Correctly Classified Observations by the New Classifier, % | Initial Classifier Accuracy, % | The Number of Correctly Classified Observations by the Initial Classifier, % |
|---|---|---|---|---|---|---|---|---|
| Neg. | 105/89 | 10/8 | 2/1 | 15/123 | 71 | 73.9 | 79 | 73.9 |
| Pos. | 10/9 | 18/11 | 5/4 | 8/23 | 78 | | 67 | |

The results of experimental studies of the algorithmic procedure for reducing the classifier for the problems being solved are shown in Tables 5 and 6.

**Table 5.** Experimental study of the classifier reduction procedure for the problem of breast cancer diagnosis.

| Optimization Problem | Set of Rules | Number of Rules | Average In-formativity | Cover of Negative Patterns | Cover of Positive Patterns | Classification Accuracy, % |
|---|---|---|---|---|---|---|
| Objective Function (1), constraint (3) | Neg. | 269 | 0.72 | 233 | 8 | 90 |
| | Pos. | 186 | 0.71 | 9 | 153 | 96 |
| Objective Function (1), constraint (3) with increasing procedure of rules | Neg. | 269 | 0.74 | 233 | 7 | 90 |
| | Pos. | 186 | 0.73 | 8 | 153 | 96 |
| Objective Function (1), constraint (3) with increasing procedure of rules and classifier reduction | Neg. | 41 | 0.79 | 243 | 5 | 90 |
| | Pos. | 36 | 0.77 | 6 | 157 | 96 |

**Table 6.** Experimental study of the classifier reduction procedure for the problem of predicting the outcome of a patient's disease with hepatitis.

| Optimization Problem | Set of Rules | Number of Rules | Average In-formativity | Cover of Negative Patterns | Cover of Positive Patterns | Classification Accuracy, % |
|---|---|---|---|---|---|---|
| Objective Function (1), constraint (3) | Neg. | 123 | 0.29 | 89 | 8 | 79 |
| | Pos. | 23 | 0.38 | 9 | 11 | 67 |
| Objective Function (1), constraint (3) with increasing procedure of rules | Neg. | 123 | 0.3 | 89 | 8 | 79 |
| | Pos. | 23 | 0.45 | 6 | 11 | 78 |
| Objective Function (1), constraint (3) with increasing procedure of rules and classifier reduction | Neg. | 71 | 0.33 | 91 | 8 | 79 |
| | Pos. | 20 | 0.47 | 6 | 11 | 78 |

As a result of applying the increasing procedure, we will obtain patterns with maximum coverage and with a higher degree, thereby increasing the reliability of the decisions made by the classifier. Improving the reliability of solutions is based on increasing the informativity of patterns, since the number of observations in one class (the coverage value for the pattern) remains unchanged, while the number of observations in another class captured by the rule decreases.

According to the results (Tables 5 and 6), the proposed procedure of classifier reduction makes it possible to increase its compactness, since the set of rules constituting it is reduced by two to six times relative to the original set of rules for a specific problem.

Table 7 compares the accuracy classification results on practical problems for the following algorithms: 1-R [34], RIPPER [35], C4.5 [35], CART [36], Adaboost [37], and a method of logical analysis of data (LAD). The classification results were obtained using the WEKA data mining system [33] for the first five algorithms and using an intelligent software system [27]. The number of observations of the examining sample is 20% for problem 1 and 15% for problem 2 of the total number of observations for the practical problems being solved. Numerous experiments were carried out for each method, and the results of the experiments were averaged.

**Table 7.** Comparison of logical classification algorithms on practical problems.

| Problem | Algorithm / Indicator | RIPPER | C4.5 | CART | Adaboost | 1-R | LAD |
|---------|------------------------|--------|------|------|----------|-----|-----|
| Problem 1 | Number of correctly classified observations, % | 92.9 | 92.9 | 92.1 | 97.4 | 88.6 | 93.2 |
| Problem 1 | Number of correctly classified observations, % | 65.2 | 56.5 | 69.5 | 69.5 | 65.2 | 73.9 |

According to the data given in Table 7, each of the algorithms presented for comparison showed high and good results in classification accuracy for the problems being solved. However, the most acceptable algorithms for these problems are LAD, Adaboost, and RIPPER. An additional feature of LAD is the ability to consider the specifics of a particular classification problem and customer requirements in its study by maintaining a balance between various criteria for comparing classification algorithms.

## 4. Conclusions

The suitability of the developed optimization models for the building of informative patterns is proved empirically. According to the obtained results, it can be noted that the optimization model for the building of partial patterns based on the boosting criterion in terms of classification accuracy is the best of the developed optimization models in this paper. It should also be noted that the patterns obtained on the basis of this optimization model are simpler and clearer.

The developed algorithmic procedures used to build the classifier have proven their effectiveness in solving practical problems. Firstly, when applying these algorithmic procedures, the classifier becomes more compact, since the number of patterns in it is reduced. Secondly, the quality of the classification of new observations is preserved.

It should be noted that the method of logical analysis of data is a flexible tool that helps, considering the specifics of a particular classification problem and the requirements of the customer (researcher) by purposefully setting its parameters. This fact is confirmed by the results of comparing the accuracy in the method of logical analysis of data with other logical classification algorithms for problems of medical prediction and diagnostics.

In this paper, only binary classification problems are considered, although the method can also be used for multiclass classification problems. The limitations of the application of the method under study in practice include the fact that it is intended only for working with binary variables. It should be noted that the procedure for binarization of nominal and quantitative features leads to the loss of part of the information. Additionally, the method is not designed to work with qualitative and fuzzy variables, which limits its use for a wide range of practical problems.

The results obtained in the paper develop the direction of logical classification algorithms and can become the basis for creating more advanced decision support systems for classifying observations. The most important advantage of such systems will be the possibility of substantiating the solution obtained during the classification. Often, as practice shows, the presence of such a possibility is primary for the user's work when solving classification problems.

As an algorithmic support for the designed decision support system, a modification of the method of logical analysis of data can be proposed. The basic idea of the modification of the method is the joint use of the developed model for the building of partial patterns based on the boosting criterion and the reduction procedure of the classifier. This modification of the method will lead to the construction of a high-precision and compact classifier for real problems of prediction and diagnostics.

The obtained results can qualitatively improve the level of solving the problems of medical prediction and diagnostics, ultimately having a positive impact on ensuring sustainable development in the healthcare sector.

In the future, it is planned to develop decision support systems for classifying observations, which will be based on the joint use of various tools that ensure the provability and

interpretability of decisions, such as logical patterns, decision trees, support vectors, and statistical classifiers. Their joint use will help a comprehensive analysis of possible solutions.

**Author Contributions:** Conceptualization, R.K. and A.S.; methodology, R.K. and A.Y.; formal analysis, R.K.; investigation, A.S. and M.P.; resources, M.P. and I.B.; data curation, R.T.; writing—original draft preparation, R.K.; writing—review and editing, I.B.; supervision, A.S. All authors have read and agreed to the published version of the manuscript.

**Funding:** The reported study was funded by RFBR, Sirius University of Science and Technology, JSC Russian Railways and Educational Fund "Talent and success", project number 20-37-51007, and by the Russian government's (9th competition) "Hybrid methods of modelling and optimization in complex systems", application number 220-8452-3649.

**Data Availability Statement:** All data are available in the public domain.

**Conflicts of Interest:** The authors declare no conflict of interest.

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
