# Peer review of "Construction of a Compact and High-Precision Classifier in the Inductive Learning Method for Prediction and Diagnostic Problems"

_information, doi:10.3390/info13120589_

Round 1

Reviewer 1 Report

After careful reading the paper, I have some suggestions.

1. Abstract: Keywords are not match, such as pattern recognition, covering and informativity. Please recheck

2. The paper is based on software simulation. Please give the block diagram and flowchart that to let the calculation steps more clearly.

3.In conclusion: Please add the limitation of your research, and do not appear Table 1-2 and Table 3-6.

4. In Reference: REF[2], REF[3], REF[6] and REF[14] are too far away, is they are the Bible in this field. Please recheck.

5. The most significant of the paper are presented:

(a)… partial patterns based on the boosting criterion is the best of the developed optimization models in this paper. 

(b)… the direction of logical classification algorithms is the basic level.

Please give some word to explain how the two methods should be used in proportion in actual example.

Author Response

Thanks for your suggestions! All explanations are in the attached document, and changes added to paper.

Reviewer 2 Report

Please see the attachment:

Author Response

Thanks for your suggestions! All changes, according to your comments, have been made to paper.

Round 2

Reviewer 1 Report

Dear Professor

About the opinions for last time, the paper had modified

1.Keyword matching

2. Ok

3. The conclusion had add the limitation.

4. But REF[2], REF[3], REF[6] and REF[14] did not replace, Could the author can present the updated references?

5. Ok.

Author Response

Thanks for your suggestions!

Reviewer 2 Report

The CV values in the numerical parts are to some strange. I ask the authors to check their correctness more preciesely. As I had mentioned in the first revision, the following recent papers about the cross validation criterion should be used for more explanations about CV definitions and applications. Otherwise, the readers may be confused with its concepts and applications.

- Optimal QR-based estimation in partially linear regression models with correlated errors using GCV criterion. Computational Statistics & Data Analysis 117 (2018), 45-61.

- Generalized cross-validation for simultaneous optimization of tuning parameters in ridge regression, Iranian Journal of Science and Technology, Transactions A: Science, 44(2020), 473–485.

Author Response

Dear Reviewer, the answer to your comment is in the attached file.
Kind regards,  Roman Kuzmich

Round 3

Reviewer 2 Report

It is clear that the aims of applying the CV criterion are different in research papers, but the concept of CV is the same. So, I propose to add the mentioned papers just for clarifying of CV definition.

Author Response

Thanks for your suggestions!
